# Consistency Beyond Contrast: Enhancing Open-Vocabulary Object Detection Robustness via Contextual Consistency Learning

## Abstract

Recent advances in open-vocabulary object detection focus primarily on two aspects: scaling up datasets and leveraging contrastive learning to align language and vision modalities. However, these approaches often neglect internal consistency within a single modality, particularly when background or environmental changes occur. This lack of consistency leads to a performance drop because the model struggles to detect the same object in different scenes, which reveals a robustness gap. To address this issue, we introduce Contextual Consistency Learning (CCL), a novel framework that integrates two key strategies: Contextual Bootstrapped Data Generation (CBDG) and Contextual Consistency Loss (CCLoss). CBDG functions as a data generation mechanism, producing images that contain the same objects across diverse backgrounds. This is essential because existing datasets alone do not support our CCL framework. The CCLoss further enforces the invariance of object features despite environmental changes, thereby improving the model's robustness in different scenes. These strategies collectively form a unified framework for ensuring contextual consistency within the same modality. Our method achieves state-of-the-art performance, surpassing previous approaches by +16.3 AP on OmniLabel and +14.9 AP on $D^3$. These results demonstrate the importance of enforcing intra-modal consistency, significantly enhancing model generalization in diverse environments. Data, code and models will be made publicly available.

## 1 Introduction

Object detection has made significant strides in recent years. However, two advanced tasks based on this technology continue to present considerable challenges: open-vocabulary object detection (OVOD) and descriptive textual object detection, such as referring expression comprehension (REC) and visual grounding (VG). Open-vocabulary object detection aims to detect previously unseen objects in dynamic environments. Recent works Dou et al. (2022); Gu et al. (2021); Li et al. (2022b); Lin et al. (2022); Minderer et al. (2023); Zhao et al. (2022); Jin et al. (2024); Zang et al. (2024) have advanced training strategies for such tasks, others Kamath et al. (2021); Kuo et al. (2022); Minderer et al. (2022); Subramanian et al. (2022) have focused on enhancing model architectures. In parallel, tasks involving referring expressions and visual grounding, which require detecting objects based on complex natural language descriptions, have shown advances in training methodologies Xie et al. (2025); Chen et al. (2025); Zong et al. (2025); Lin et al. (2024); Peng et al. (2023), architectural improvements Yin et al. (2025); Lin et al. (2023); You et al. (2023) and the use of the capabilities of large models Shen et al. (2025); Xuan et al. (2024); Zhan et al. (2024).

Despite these advancements, there is still a crucial gap in addressing the internal consistency within each input image and query. We identify an issue in existing models Dou et al. (2022); Li et al. (2022b; 2023a): the features of the same object tend to vary significantly across different scenes, which indicates that current models may overfit to specific training backgrounds. This inconsistency not only affects the detection stability but might also degrade the model's generalization ability, raising an important question: *Can we obtain object features that are robust to environmental changes?* To validate this, we construct the $D^3_{BC}$ test set by applying background replacement to the original $D^3$ dataset. Detailed in Section 4.3, baseline methods suffer notable performance drops under this setting, highlighting their limited robustness to contextual changes. In contrast, our method maintains

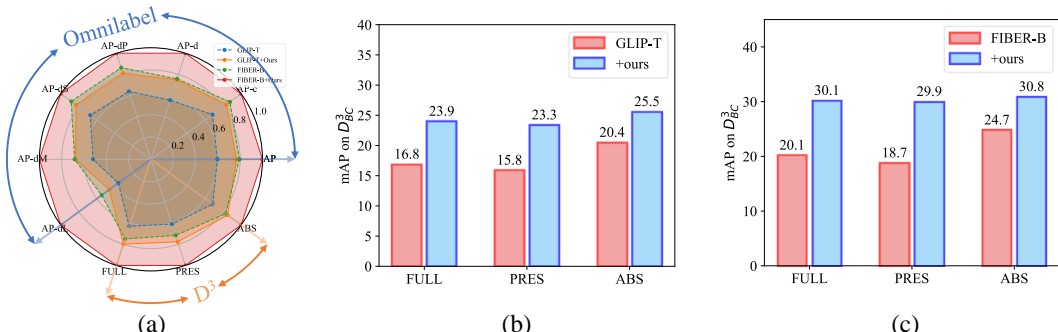

(a)          (b)          (c)

Figure 1: Performance and robustness comparison of different methods. (a) Our approach, with Contextual Consistency Learning, achieves the best overall results, reaching a normalized score of 1 in all metrics. (b,c) Benchmark backgrounds are altered to test robustness. Tested on $D^3_{\text{BC}}$, baseline methods degrade, while ours remains stable. See Section 4.3 for details.

performance comparable to that on the original benchmark. As shown in Figure 1, our experimental results demonstrate that addressing this issue significantly improves model performance.

To address this issue, we propose the CCL framework that enforces invariance of object features across different scenes, as shown in Figure 2. However, existing datasets exhibit a notable limitation: they lack comprehensive data pairs that depict the same object in diverse contextual settings. This data gap is crucial because CCL requires models to encounter and learn from variations of the same object in different environments or scenarios. Without such diverse representations, models struggle to generalize under varying real-world conditions. To overcome this limitation, we introduce CBDG, which first increases the number of categories and then leverages SAM Kirillov et al. (2023) and the Stable Diffusion model Rombach et al. (2022) to generate data pairs across different scenes while ensuring consistent foreground objects, thus improving both category variation and background diversity in training data.

Our experimental results demonstrate significant improvements on two challenging benchmarks, $D^3$ Xie et al. (2023) and OmniLabel Schulter et al. (2023), achieving +16.3 AP on OmniLabel and +14.9 AP on $D^3$. The proposed CBDG and CCLoss are complementary components that collectively form a robust training paradigm. Specifically, the CBDG improves feature learning through diverse scene-object compositions, while the CCLoss ensures robust feature representation across varying backgrounds. Furthermore, our approach is fundamentally model-agnostic, enabling seamless integration into a wide range of existing architectures, such as Dou et al. (2022); Li et al. (2022b), with consistent performance gains across different frameworks.

In summary, the contributions are as follows.

- This study identifies an issue where object features are highly susceptible to environmental changes, leading to potential overfitting and poor generalization to unseen scenarios.

- To ensure feature robustness to context changes, we propose CCL, which enforces object consistency across backgrounds via CBDG and CCLoss.

- Our method is simple, efficient, and model-agnostic, imposing no additional inference overhead while consistently delivering performance improvements across diverse datasets and models. Moreover, despite working with a much smaller subset of the original dataset, we achieve state-of-the-art results on two descriptive open-vocabulary detection benchmarks.

## 2 RELATED WORK

**Vision language localization tasks.** Open-vocabulary object detection (OVOD) aims to enable models to recognize novel objects or unseen categories during inference Gu et al. (2021); Minderer et al. (2023); Zareian et al. (2021); Du et al. (2022) , extending beyond traditional categorical detection. However, this ability is typically limited to detecting object categories based on labels, rather than understanding long descriptions. In contrast, referring expression comprehension (REC)

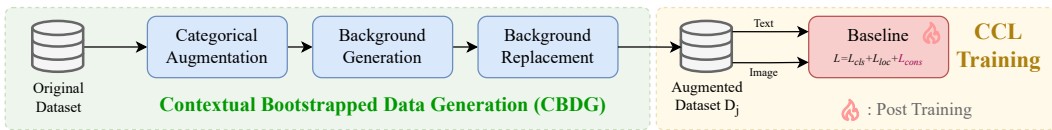

Figure 2: Overview of our approach. CBDG generates $D_j$ via Categorical Augmentation, Background Generation and Background Replacement. CCL training uses $D_j$ with CCLoss added to total loss.

involves understanding and localizing objects in an image based on natural language descriptions that refer to specific instances of objects Yu et al. (2016); Wu et al. (2020); Mao et al. (2016), making it inherently more flexible and context-aware. While OVOD and REC both address the challenge of understanding objects in images, we focus on simultaneously handling novel categories and complex natural language descriptions. We opt for described object detection (DOD) Xie et al. (2023) and OmniLabel Schulter et al. (2023) as robust solutions to these challenges, as they incorporate both the recognition of novel categories and the understanding of intricate descriptions.

**Diffusion models for scenario generation.** Stable Diffusion Rombach et al. (2022) marks a shift in text-to-image synthesis by operating in a compressed latent space using iterative denoising. Unlike GANs Goodfellow et al. (2014); Mirza & Osindero (2014) or VAEs Kingma et al. (2013); Van Den Oord et al. (2017) that generate images in pixel space, it uses a VAE to encode images into low-dimensional latents, allowing efficient training and high-resolution output. Guided by a pre-trained CLIP text encoder, the model aligns generated images with complex textual descriptions, from concrete objects to abstract scenes.

Recent diffusion-based methods have shown strong performance in image inpainting, enabling object and scene editing via text or spatial inputs. GLIDE Nichol et al. (2021) enables text-guided object replacement while preserving scene consistency, and GLIGEN Li et al. (2023b) extends this by incorporating bounding boxes for more precise control over object placement. For background replacement, IAM Yu et al. (2023) integrates segmentation with diffusion to regenerate regions based on textual prompts. Despite their effectiveness, these methods often suffer from boundary artifacts due to over-smoothing during denoising. We evaluate GLIDE, IAM, and Stable Diffusion Rombach et al. (2022) in CBDG and ultimately choose Stable Diffusion for background generation.

**Cross-modal object detection models.** With the advancement of multimodal vision language models, such as CLIP Radford et al. (2021) and ALIGN Jia et al. (2021), the development of methods that integrate vision and language to address visual recognition tasks has emerged as a prominent trend. GLIP Li et al. (2022b), based on CLIP Radford et al. (2021), leverages free-form language supervision during training and frames object detection as visual localization, constructing a foundation for semantically enriched pre-trained models. Building on this, FIBER Dou et al. (2022) employs a two-stage training approach, transitioning from coarse-grained to fine-grained, enhancing the adaptability of the pre-trained model to a broad spectrum of downstream tasks at both image-level and region-level. In our work, we use GLIP Li et al. (2022b) and FIBER Dou et al. (2022) as baseline models and incorporate our CCL method to validate the experimental results.

## 3 METHOD

### 3.1 OVERVIEW

We introduce CCL, a novel framework designed to address the challenge of maintaining detection and grounding consistency when models encounter diverse and unseen object categories across varying contextual backgrounds. To achieve this goal, we address two fundamental aspects of the problem: the lack of appropriate training data and the need for effective consistency-preserving mechanisms.

In Section 3.2, we describe our CBDG pipeline, which leverages advanced segmentation and generative models to create a rich and varied dataset. This data preparation process is specifically designed to support our consistency learning objectives. Following this, in Section 3.3, we detail our CCLoss formulation, which ensures that the model learns to maintain object identity across different backgrounds.

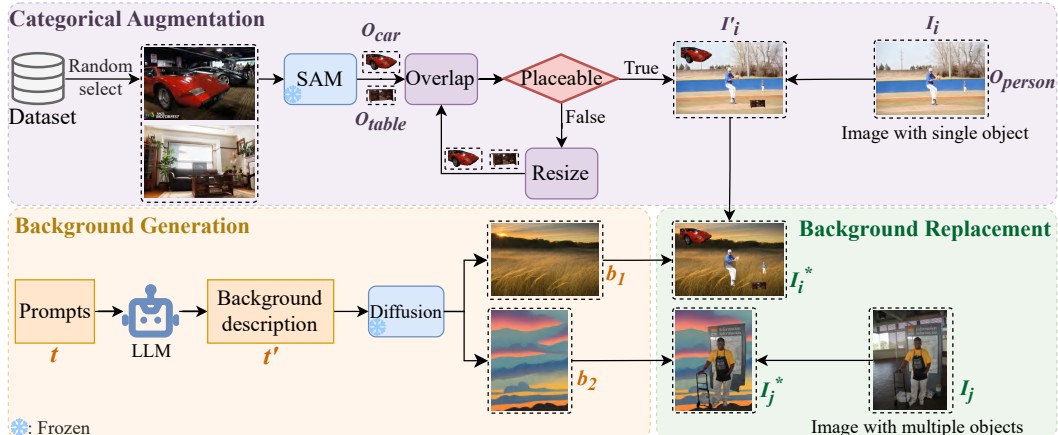

Figure 3: CBDG Pipeline. We use ChatGPT to generate background prompts for a diffusion model, enabling diverse background synthesis. For single-class images, CBDG augments object categories before background replacement. For multi-class images, CBDG replaces only the background.

## 3.2 CONTEXTUAL BOOTSTRAPPED DATA GENERATION

Current open-set visual grounding methods struggle to maintain robustness across diverse real-world scenarios, particularly when objects appear in unfamiliar contextual settings. This limitation stems from a fundamental data scarcity: existing datasets rarely capture the full spectrum of object-background interactions, leading to biased model performance. To overcome this, we propose a multistage data augmentation framework that synthesizes diverse and realistic object-context compositions by combining SAM-based object manipulation with text-guided background generation, as shown in Figure 3. Our method constructs a compositionally diverse joint dataset $D_j$ that mitigates common inpainting artifacts and improves model generalization.

**Categorical augmentation.** In our approach, we use the Flickr30k Entities visual grounding dataset Plummer et al. (2015) alongside a subset of the Objects365 object detection dataset Shao et al. (2019) to create a combined training dataset. The selected subset of Objects365 tends to contain images dominated by a few categories, with multiple instances of that category present. Details are discussed in Supplementary Section D.3 and Section D.5.

For images with a single object, we aim to enhance the diversity of object categories within each image by introducing objects from different categories while maintaining spatial and contextual coherence. To achieve this, we leverage the SAM model Kirillov et al. (2023) to extract precise objects $O_i$ and the corresponding position $(x_o, y_o)$ for each image $I$, where $i$ means the category ID to which the object belongs. The object masks allow us to identify individual objects and their spatial locations. Based on these masks, we randomly select objects $O_{i \notin C}$ from other images within the same subset but belonging to different categories, where $C$ represents the category set. Then these objects are positioned in the current image at carefully chosen locations. Specifically, we define $P = \{(x_1, y_1), (x_2, y_2), ..., (x_N, y_N)\}$ the potential placement position set for the new object, $N$ is the number of candidate locations. From these positions, we randomly select $(x, y) \in P \backslash (x_o, y_o)$ that does not overlap with the existing objects in the image, ensuring a clean and non-interfering insertion of the new object. This process of placement can be formalized as:

$$\texttt{Augmentation}: (x_{o_k}, y_{o_k}) \in P \backslash (x_o, y_o), o_k \in O_{i \notin C}, \tag{1}$$

where $k$ represents the category of selected object, $(x_{o_k}, y_{o_k})$ denotes the position randomly chosen from the set of candidate locations according to the above rule. After categorical augmentation, the original image $I$ becomes $I'$.

In scenarios where no suitable empty position is available, such as when the current image contains large objects or a large number of dispersed objects, which results in limited available space, we adopt a resizing strategy. In these cases, we reduce the size of the new object to $1/\alpha$ of its original size and attempt to place it again, where $\alpha$ is a scaling factor. This process is repeated until an empty

Figure 4: Four groups of images are shown, each composed of four sub-images: the leftmost sub-image in every group is the original, while the remaining three display background replacements.

placement area is found or the number of resizing attempts exceeds a threshold $N_R$. If resizing attempts fail to find a suitable location, we abandon the current image and instead select another image to enhance the diversity of object categories, thus ensuring a broader range of category representation in the final dataset.

**Background generation.** With more object categories added, the foreground dataset now includes images with varied labels and their corresponding bounding boxes. To reduce model overfitting and improve generalization, we next generate diverse background images, placing the same objects in different scenes. Image inpainting methods Nichol et al. (2021); Li et al. (2023b); Yu et al. (2023) often struggle with blurred edges and backgrounds that still reflect foreground features, making realistic scene changes difficult (see Supplementary Section B.6). To avoid these issues, we use a simpler alternative that better separates foreground from background.

Instead of relying on limited original image content, we generate new and simple backgrounds directly. Using a Large Language Model (LLM) Brown et al. (2020), denoted as $\mathcal{G}$, we create text prompts in three categories: *Seasonal, Sky, and Natural Landscape*, to ensure variety and relevance. Details of these prompts are provided in Supplementary Section D.1. These prompts are input into Stable Diffusion $\mathcal{D}$ Rombach et al. (2022), which generates matching background images. This method allows us to build a diverse, context-aware background dataset without the limitations of inpainting:

$$\text{Generation}: b = \mathcal{D}(t'), t' = \mathcal{G}(t), \tag{2}$$

where $t \in \{$*Seasonal, Sky, Natural Landscape*$\}$, $t'$ is the background description generated by ChatGPT, and $b$ represents the background image generated by stable Diffusion, which constitutes the background dataset $D_{bg}$. We further analyze the diversity of generated scenes in Supplementary Section B.4 and Section D.4.

**Background replacement.** At this stage, we have both the generated background dataset $D_{bg}$ and the foreground dataset $D_{fg}$ with various object categories. To create new image variations, we randomly select background images for each foreground image. Using the bounding boxes, we extract foreground objects with the SAM model $\mathcal{S}$ Kirillov et al. (2023). The post-processing techniques are detailed in Supplementary Section D.2. These objects and their spatial layout are kept unchanged. After isolating the foreground, we replace the original background with a selected one, generating multiple new images per original. The foreground stays the same, while the backgrounds vary, producing diverse scenes with consistent object content. The replacement process is defined as:

$$\text{Replacement}: I^* = \mathcal{S}(I', bbox) \oplus b, b \in D_{bg}, \tag{3}$$

where $bbox$ represents the bounding boxes of objects in the image $I'$ after categorical augmentation, $\oplus$ denotes the composition of foreground and background, $I^*$ represents the image with replaced background.

CBDG enables us to significantly augment the dataset with diverse background settings while maintaining the integrity of the foreground objects, providing a more robust foundation for training our model. As shown in Figure 4, after CBDG, for each original image, several additional images are generated with replaced backgrounds. This results in a total of $K$ images per original, all sharing the same foreground objects, but differing in their backgrounds. $K$ represents the batch size used during training. Alternative data generation schemes are also compared, see Supplementary Section B.3. The augmented images are then utilized for the subsequent consistency constraints in our approach.

Figure 5: CCL Framework. Visual and textual features are encoded, with regional features pooled into CAAF. Consistency loss is applied within each modality.

## 3.3 CONTEXTUAL CONSISTENCY LOSS

Given that after CBDG, we now have access to a dataset $D_j$ in which each group of images shares the same foreground object but varies in background. We introduce the Contextual Consistency Loss (CCLoss), a novel training objective designed to enforce representation invariance for the same object category across varying contextual environments. As illustrated in Figure 5, our method uses CCLoss to maintain the consistency of foreground object representations across different backgrounds. By constructing training batches that contain instances of the same object under different contextual settings, CCLoss encourages the model to focus on semantically meaningful foreground features rather than background-dependent or spurious cues. This section elaborates on the underlying model architecture, the detailed formulation of consistency loss, and its integration into the overall training objective.

**Model architecture.** We employ a language-based object detector Li et al. (2022b); Dou et al. (2022) as the backbone for feature extraction and object detection, taking advantage of its strong capability in bridging vision and language representations. Specifically, images and textual descriptions are first encoded to obtain their respective feature embeddings, ensuring a comprehensive understanding of both modalities. These extracted features are subsequently processed through a Feature Pyramid Network (FPN), which effectively refines and integrates multiscale representations, thereby enhancing detection performance across various object sizes and contexts. To further improve localization accuracy, the refined image features are then passed on to DynamicHead, a dedicated module designed to predict a set of candidate regions where objects are most likely to be located. This hierarchical and adaptive processing pipeline ensures robust and efficient object detection.

**Consistency loss.** During the training phase, we organize each batch by grouping images that share identical foreground objects but exhibit diverse background settings. This arrangement enables the computation of the CCLoss function, which serves as a critical mechanism for training the model to preserve invariant representations of foreground objects across varying contextual environments.

The total loss function, as depicted in Eq. 4, comprises three fundamental components: localization loss, classification loss, and contextual consistency loss ($\mathcal{L}_{\text{cons}}$). Each of these components contributes to optimizing the model performance in different aspects: precise object localization, accurate category classification, and robust feature representation that maintains foreground consistency irrespective of background variations. The first two components of the loss function are detailed in GLIP Li et al. (2022b). The integration of these loss terms ensures a balanced optimization process that addresses discriminative and invariant feature learning.

$$\mathcal{L} = \mathcal{L}_{\text{cls}} + \mathcal{L}_{\text{loc}} + \mathcal{L}_{\text{cons}}, \tag{4}$$

Eq. 5 provides the formulation of CCLoss. CCLoss combines the text and image modality losses with weighting factors. $\lambda_{\text{T}}$ and $\lambda_{\text{I}}$ are weighting parameters to balance the loss contributions in the text and the image modality.

$$\mathcal{L}_{\text{cons}} = \lambda_{\text{T}} \cdot \mathcal{L}_{\text{T}} + \lambda_{\text{I}} \cdot \mathcal{L}_{\text{I}}, \tag{5}$$

For image features obtained from the image encoder, we first perform a pooling operation on them to obtain the Context-Aware Aggregated Feature (CAAF), denoted as $f$, followed by applying a

consistency loss among the CAAF. Given a batch with $C$ categories and $K$ images, the contrastive loss for the vision modality is defined as:

$$\mathcal{L}_{\mathrm{I}} = -\frac{1}{CK} \sum_{c=1}^{C} \sum_{k=1}^{K} \log \frac{\exp\big(\mathrm{sim}(\mathbf{f}_{ck}, \mathbf{f}_c)/\tau\big)}{\sum\limits_{c'=1}^{C} \sum\limits_{k'=1}^{K} \exp\big(\mathrm{sim}(\mathbf{f}_{ck}, \mathbf{f}_{c'k'})/\tau\big)}, \tag{6}$$

where $\mathbf{f}_{ck}$ is the $k$-th image feature of the $c$-th category. $\mathbf{f}_{c'k'}$ is the $k'$-th image feature of the $c'$-th category. $\mathbf{f}_c$ is the centroid of the image features for the $c$-th category, calculated as the mean of the $K$ image features. $\mathrm{sim}(\cdot, \cdot)$ is cosine similarity. $\tau$ is the temperature parameter.

Similarly, for text features, we implement a contrastive learning objective that promotes feature clustering within the same category while enforcing separation among different categories. However, the application of this text contrastive loss is contingent upon the baseline architecture: When using FIBER as the baseline, where cross-modal interactions between image and text encoders are enabled, we fully utilize this loss term. In contrast, when employing GLIP as the baseline, which processes image and text modalities independently, we effectively disable this component by setting its weight $\lambda_{\mathrm{T}}$ to zero. Given a batch with $C$ categories and $K$ images, the contrastive loss for the text modality is defined as:

$$\mathcal{L}_{\mathrm{T}} = -\frac{1}{CK} \sum_{c=1}^{C} \sum_{k=1}^{K} \log \frac{\exp\big(\mathrm{sim}(\mathbf{t}_{ck}, \mathbf{t}_c)/\tau\big)}{\sum\limits_{c'=1}^{C} \sum\limits_{k'=1}^{K} \exp\big(\mathrm{sim}(\mathbf{t}_{ck}, \mathbf{t}_{c'k'})/\tau\big)}, \tag{7}$$

where $\mathbf{t}_{ck}$ is the $k$-th text feature of the $c$-th category. $\mathbf{t}_{c'k'}$ is the $k'$-th text feature of the $c'$-th category. $\mathbf{t}_c$ is the centroid of the text features for the $c$-th category, calculated as the mean of the $K$ text features. The design of our CCLoss follows a progressive evolution, with the detailed process provided in Supplementary Section B.7.

## 4 EXPERIMENTS

### 4.1 EXPERIMENTAL DESIGN

**Training setup.** To evaluate the generalizability of our proposed method, we use two baseline models, GLIP Li et al. (2022b) and FIBER Dou et al. (2022). These models serve as benchmarks for comparison. The datasets used to train the baseline models are 1) Objects365 (O365) Shao et al. (2019) and 2) GoldG, including Flickr30K Plummer et al. (2015), VG Caption Krishna et al. (2017), and GQA Hudson & Manning (2019), which together contain 0.8 million images, providing a diverse and large-scale training set.

In contrast, for our method, we work with a smaller subset of the original dataset, with only 0.25 million images as the initial joint dataset for CBDG. Specifically, we incorporate the Flickr30k Entities Plummer et al. (2015) dataset along with only 0.22M images of the Objects365 dataset Shao et al. (2019), which is much smaller than the full dataset used for the baselines.

We generate three main categories of background images in CBDG: seasonal, sky, and natural landscape. In total, we have 13,185 unique descriptions, resulting in 144,654 generated images. The breakdown of categories and the corresponding number of images is as follows: seasonal (3387 descriptions, 48,156 images), sky (3399 descriptions, 48,210 images), and natural landscape (3399 descriptions, 48,288 images).

For training, we use publicly available pre-trained model checkpoints of both GLIP Li et al. (2022b) and FIBER Dou et al. (2022). These pre-trained weights serve as the starting point for fine-tuning. We fine-tune the model for one epoch on our dataset $D_j$. After this fine-tuning process, we obtain the final results, which demonstrate the effectiveness of our method when applied to a smaller and more constrained dataset. The implementation details and computational cost can be found in Supplementary Section A. We report the choice and tuning of hyperparameters in Supplementary Section B.1.

**Benchmark selection.** We choose OmniLabel Schulter et al. (2023) and $D^3$ Xie et al. (2023) as benchmark evaluation methods, both of which use Average Precision (AP) as the evaluation

Table 1: Performance of our method compared with SOTA methods.

| Method | OmniLabel | | | | | | | $D^3$ | | |
|---|---|---|---|---|---|---|---|---|---|---|
| | AP | AP-c | AP-d | AP-dP | AP-dS | AP-dM | AP-dL | FULL | PRES | ABS |
| Detic Zhou et al. (2022) | 8.0 | 15.6 | 5.4 | 8.0 | 5.7 | 5.4 | 6.2 | - | - | - |
| OFA-DOD Xie et al. (2023) | - | - | - | - | - | - | - | 21.6 | 23.7 | 15.4 |
| RelationLLM-L Xie et al. (2025) | - | - | - | - | - | - | - | 24.3 | 24.6 | 23.4 |
| GN-GLIP Zhao et al. (2024) | 22.2 | 27.2 | 18.8 | 29.0 | - | - | - | 21.4 | 20.6 | 23.7 |
| GN-FIBER Zhao et al. (2024) | 28.1 | 32.1 | 25.1 | 36.5 | - | - | - | 26.0 | 25.2 | 28.1 |
| ROD-MLLM Yin et al. (2025) | - | - | 25.3 | 30.9 | 31.8 | 24.5 | 21.0 | 29.7 | 30.0 | 28.7 |
| Real-Model Chen et al. (2025) | - | - | 36.5 | **52.1** | **54.4** | 33.2 | 25.5 | 34.1 | 34.4 | 33.2 |
| GLIP-T Li et al. (2022b) | 19.3 | 23.6 | 16.4 | 25.8 | 29.4 | 14.8 | 8.2 | 19.1 | 18.3 | 21.5 |
| +ours | 32.2 | 36.1 | 28.8 | 39.8 | 43.3 | 26.5 | 17.6 | 30.0 | 29.2 | 32.3 |
| FIBER-B Dou et al. (2022) | 25.7 | 30.3 | 22.3 | 34.8 | 38.6 | 19.5 | 12.4 | 22.7 | 21.5 | 26.0 |
| +ours | **42.0** | **44.1** | **39.2** | 50.8 | 53.7 | **38.2** | **32.3** | **37.6** | **37.2** | **38.8** |

metric. The reason we select these two benchmarks is that they not only provide object category labels but also include a rich diversity of textual descriptions, which place a greater emphasis on the model's ability to understand and interpret language. This aspect makes these benchmarks particularly valuable for evaluating the model's performance in tasks involving both visual and linguistic information. Compared to other REC Yu et al. (2016); Wu et al. (2020); Mao et al. (2016) and OVOD Gupta et al. (2019); Chen et al. (2015); Krasin et al. (2017) benchmarks, $D^3$ Xie et al. (2023) and OmniLabel Schulter et al. (2023) offer a broader evaluation of object detection capabilities. These benchmarks include negative samples and more precisely defined bounding boxes corresponding to textual descriptions, which can refer to zero, one, or multiple objects in the image. This makes the tasks more challenging and forces the model to effectively localize and recognize objects based on a range of different descriptions and contexts, offering a more comprehensive test of its generalization and performance in diverse scenarios.

## 4.2 COMPARISON WITH SOTA METHODS

Table 1 presents a comparison between our method and the current SOTA methods on the Omni-Label Schulter et al. (2023) and $D^3$ Xie et al. (2023) benchmarks. The first column lists various model methods, followed by seven columns representing the seven AP metrics on OmniLabel. These metrics include: plain categories (AP-c) and free-form descriptions (AP-d). AP-dP evaluates only positive descriptions. AP-dS/M/L assess descriptions of varying lengths (up to 3 words, 4-8 words, and more than 8 words). The last three columns represent the AP metrics on $D^3$: FULL, PRES, and ABS, which evaluate all descriptions, only presence descriptions, and only absence descriptions, respectively.

We use GLIP-T Li et al. (2022b) and FIBER-B Dou et al. (2022) as baselines and fine-tune them on our method. With the integration of our proposed CCL method, significant improvements are observed across multiple benchmarks. Specifically, when applied to the FIBER baseline, the method achieves a notable increase of +16.3 AP on the OmniLabel benchmark and +14.9 AP on the $D^3$ benchmark. Similarly, when implemented with the GLIP baseline, our method demonstrates consistent performance gains, achieving +12.9 AP on the OmniLabel benchmark and +10.9 AP on the $D^3$ benchmark. These results underscore the effectiveness of our approach in improving contextual understanding and consistency across diverse datasets. We further evaluate our method on phrase grounding tasks to demonstrate broader applicability (see Supplementary Section B.5).

## 4.3 ROBUSTNESS EVALUATION UNDER BACKGROUND VARIATIONS

To quantitatively assess the robustness of open-vocabulary detection (OVD) models under environmental and background variations, we introduce a new experiment setup derived from the $D^3$ dataset. For each of the 10,578 original images in $D^3$, we generate three additional variants by replacing the background using the CBDG method proposed in this work. These new background images are generated independently of the training data, ensuring no overlap or information leakage. The result-

Table 2: Performance comparison on $D^3{}_{\mathrm{BC}}$ benchmark across different models and settings.

| Method | $D^3{}_{\mathrm{BC}}$ | | |
| --- | --- | --- | --- |
| | FULL | PRES | ABS |
| GLIP-T | 16.8 | 15.8 | 20.4 |
| +ours | 29.6 | 28.9 | 31.9 |
| FIBER-B | 20.1 | 18.7 | 24.7 |
| +ours | 33.1 | 32.8 | 34.0 |

Table 3: Ablation study of contextual bootstrapped data generation and CCLoss.

| Method | OmniLabel | | | $D^3$ | | |
| --- | --- | --- | --- | --- | --- | --- |
| | AP | AP-c | AP-d | FULL | PRES | ABS |
| GLIP-T | 19.3 | 23.6 | 16.4 | 19.1 | 18.3 | 21.5 |
| +data | 24.8 | 29.2 | 21.8 | 23.2 | 22.5 | 25.3 |
| +ours | 32.2 | 36.1 | 28.8 | 30.0 | 29.2 | 32.3 |
| FIBER-B | 25.7 | 30.3 | 22.3 | 22.7 | 21.5 | 26.0 |
| +data | 32.7 | 35.8 | 29.6 | 29.1 | 28.3 | 31.2 |
| +ours | 42.0 | 44.1 | 39.2 | 37.6 | 37.2 | 38.8 |

ing dataset, termed $D^3{}_{\mathrm{BC}}$, consists of the original images and their background-altered counterparts, totaling 42,312 samples. We evaluate two representative baseline models, GLIP-T and FIBER-B, on both $D^3$ and $D^3{}_{\mathrm{BC}}$, and further examine their performance when enhanced with our proposed CCL method. This yields four experimental settings. As summarized in Table 2, both baselines exhibit substantial performance degradation on $D^3{}_{\mathrm{BC}}$, revealing their susceptibility to background shifts. However, models incorporating our CCL approach demonstrate significantly improved robustness with much smaller performance drops. These results highlight the effectiveness of CCL in improving model resilience to environmental variations. Moreover, this experiment suggests that our method maintains robustness not only under background shifts but also across different domains.

## 4.4 Ablation Study on CBDG and CCLoss

Given that our method is fundamentally grounded in consistency and incorporates a certain degree of data generation, we perform a series of ablation experiments to evaluate the contribution of each individual component. In particular, we conduct two distinct experimental setups to assess the impact of both CBDG and CCLoss. The first experiment introduces CBDG to the baseline model, followed by fine-tuning the model for one epoch on $D_j$. To ensure a fair comparison, we keep the training parameters consistent with those used in the baseline experiment. The second experiment represents our complete experimental setup, adding CBDG and CCLoss to the baseline model and fine-tuning the model for one epoch. As shown in Table 3, both CBDG and CCLoss play an essential role in enhancing the model's performance. CBDG increases the diversity of training data, improving the model's robustness across varying conditions. Meanwhile, the CCLoss reinforces object consistency across different contexts, ensuring that the model can reliably detect and localize objects regardless of their surrounding environment. The combined effects of these two components contribute significantly to the observed performance improvements. We further analyze the impact of dataset scale in Supplementary Section B.2.

## 5 Conclusion

**Summary.** We propose Contextual Consistency Learning (CCL) to tackle inconsistent object feature representation in descriptive open-vocabulary object detection. CCL combines Contextual Bootstrapped Data Generation (CBDG) and Contextual Consistency Loss (CCLoss). CBDG uses SAM and Stable Diffusion to generate diverse scene-object compositions, while CCLoss enforces feature invariance across backgrounds. Despite using significantly less data, CCL improves model performance. It is model-agnostic, incurs no inference overhead, and integrates easily into existing architectures. Our work underscores the importance of intra-modal consistency for robust object detection in dynamic environments, paving the way for future extensions to broader vision-language tasks and large-scale models.

**Limitation & Future work.** Due to the inherent limitations of SAM, segmentation errors or under-segmentation may occur when extracting foreground objects in our CBDG. Although our post-processing techniques effectively mitigate these issues and achieve SOTA performance, further research is needed to completely eliminate such problems and further enhance performance.

## ETHICS STATEMENT

This work does not involve human subjects, private or sensitive data, or applications that may cause direct harm. All datasets used in this study are publicly available and widely adopted in the research community. Our method focuses on improving robustness in object detection by generating synthetic background variations, which does not introduce new ethical concerns. We have taken care to avoid reinforcing social biases, and the proposed framework is intended solely for academic research.

## REPRODUCIBILITY STATEMENT

We have made significant efforts to ensure the reproducibility of our results. All implementation details, including network architectures, training schedules, and hyperparameters, are described in the main paper and supplement. The datasets used are publicly available, and we provide detailed descriptions of preprocessing steps in the supplementary materials. Furthermore, the theoretical formulation of our loss function is fully detailed in Supplementary Section B.7, with proofs and additional derivations provided in the supplement. As mentioned in the abstract, data, code and models will be made publicly available.

## USE OF LARGE LANGUAGE MODELS (LLMS)

We use large language models (LLMs) solely to assist with the polishing of English writing, such as improving grammar, clarity, and readability. In addition, using a Large Language Model (LLM) Brown et al. (2020), we generate prompts that are subsequently fed into Stable Diffusion Rombach et al. (2022) to synthesize background images for our experiments. No part of the research design, experimental implementation, data analysis, or result interpretation relied on LLMs. All scientific contributions, ideas, and experiments are conceived and conducted entirely by the authors.

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
