This document supplements the main paper as follows.

A. Details

- Sect. A.1 presents the implementation details and computational cost.

B. Additional Experiments

- Sect. B.1 presents the choice of hyperparameters.
- Sect. B.2 presents the analysis of post-training data scale.
- Sect. B.3 presents the choice of generation scheme.
- Sect. B.4 presents the analysis on CBDG scene diversity.
- Sect. B.5 presents additional experiments on phrase grounding.
- Sect. B.6 presents an analysis of inpainting methods for background change.
- Sect. B.7 presents the progressive design of the intra-modal consistency loss.

C. Related Work

- Sect. C.1 presents the related work of robust object detection.
- Sect. C.2 presents the related work of data augmentation for detectors.
- Sect. C.3 presents the related work of cross-modal object detection models.
- Sect. C.4 presents the survey of existing background datasets.

D. CBDG Further Analysis

- Sect. D.1 presents the prompts used for background generation with LLM.
- Sect. D.2 presents the post-processing techniques during background replacement stage of CBDG.
- Sect. D.3 provides an analysis of the processing procedures for different datasets.
- Sect. D.4 presents the analysis on background generation.
- Sect. D.5 provides an analysis of image selection and category augmentation.
- Sect. D.6 presents the pseudocode of the CBDG pipeline.

# A  DETAILS

## A.1  IMPLEMENTATION DETAILS AND COMPUTATIONAL COST

Our CCL framework consists of two main components: CBDG and training with CCLoss. The experimental details are as follows:

For Categories Augmentation, we select $P$ potential object placement locations, setting the number of positions $N = 100$. If the number of available placement locations is fewer than 5, the object's width and height are scaled down by a factor of $1/\alpha$, where $\alpha = 2$. If the number of scaling operations exceeds a threshold $N_R$, we switch to another object, with $N_R = 2$.

For model fine-tuning, we initialize GLIP and FIBER with their official checkpoints, which are trained on large-scale and diverse datasets such as GoldG and Objects365. We then fine-tune these models on our augmented dataset, generated via the CBDG pipeline. To ensure the models can adapt effectively to our domain-shifted data, we reset the learning rate to 1e-4 and use the Adam optimizer. This relatively high learning rate facilitates optimization toward new minima that are better aligned with our objective of contextual robustness, rather than preserving the prior distribution learned from the baseline training set. Both baseline models are fine-tuned with a batch size of 4, while all other training hyperparameters remain unchanged. The weight settings for CCLoss during training are as follows: When using FIBER as the baseline, $\lambda_T$ is set to 0.05, and $\lambda_I$ is set to 0.15. When using GLIP as the baseline, $\lambda_T$ is set to 0, and $\lambda_I$ is set to 0.15. The temperature parameter $\tau$ in the loss function is set to 1 by default.

While our method introduces an additional data generation step (CBDG), it is important to note that our overall training efficiency remains significantly higher than that of baseline methods. Within the CBDG stage, the most time-consuming part is the background generation step. We generated a total of 144,654 unique background images, which takes about 101 hours on 4 A100 GPUs. However, this step is conducted only once, and the generated data is reused across models and experiments. Other steps in CBDG (e.g., foreground segmentation and compositing) are relatively fast and negligible in cost.

The CCL post-training stage is lightweight. On 4 NVIDIA A100-SXM4-40G GPUs, it takes approximately 7 hours for FIBER and 9 hours for GLIP. This is significantly lower than the 540 GPU-hours reported for FIBER's original fine-grained pretraining. In our case, we only perform a single-epoch fine-tuning, which highlights the data efficiency enabled by our framework.

The following results are measured with an input image resolution of 224×224. When using GLIP as the baseline, the model has 54.81 GFLOPs and 195.19M parameters; when using FIBER as the baseline, it has 51.98 GFLOPs and 200.02M parameters. In both cases, our method introduces less than 1% additional FLOPs and parameters relative to the baseline, resulting in negligible overhead. This demonstrates that our framework preserves computational efficiency while delivering consistent performance gains.

# B  EXPERIMENT

## B.1  CHOICES OF HYPERPARAMETERS

In our experiments, we perform systematic tuning on the FIBER baseline to optimize the performance of the proposed CCLoss. Our tuning procedure is conducted in two stages. First, we fix the temperature parameter $\tau$ to its commonly used default value of 1.0, following prior contrastive learning literature, such as in CLIP and SimCLR. This choice has been widely validated to provide a good balance between hard and easy negatives without introducing instability in the optimization process. Moreover, we find that performance is relatively insensitive to minor variations in $\tau$, so we opt to retain the default value and focus our tuning efforts on the loss weight parameters.

As shown in Table 4, we begin by tuning $\lambda_I$, the weight of loss of consistency on the image side. We vary this parameter while holding $\lambda_T = \lambda_I$, and find that a value around 0.15 yields the best trade-off between stability and performance gain. Once $\lambda_I$ is fixed to 0.15, we then tune $\lambda_T$, the weight of loss of consistency on the text side. A value near 0.05 proves to be most effective when applied with the FIBER baseline, which supports cross-modal interactions. Note that for the GLIP baseline, we set the same $\lambda_I = 0.15$ and set $\lambda_T=0$ since GLIP decouples visual and textual modalities, rendering this loss component ineffective in that context.

Table 4: Tuning on the FIBER baseline

| Tuning | $\lambda_I$ | $\lambda_T$ | OmniLabel (AP) | $D^3$ (FULL) |
|--------|-------------|-------------|----------------|--------------|
| 0 | 0 | 0 | 32.7 | 29.2 |
| 1 | 0.5 | 0.5 | 35.8 | 31.5 |
| 2 | 0.2 | 0.2 | 38.2 | 34.0 |
| 3 | 0.1 | 0.1 | 39.9 | 36.1 |
| 4 | 0.15 | 0.15 | 41.2 | 37.1 |
| 5 | 0.15 | 0.1 | 41.7 | 37.4 |
| 6 | 0.15 | 0.05 | 42.0 | 37.6 |

## B.2  ANALYSIS ON DATA SCALE

To further explore the impact of data scale, we conduct an additional experiment where we systematically modify the size of $D_j$. In this experiment, we reduce the dataset size and evaluate the model's performance to determine how much the consistency-based approach can still contribute under these conditions. The reduction scale factor is $K_r$. As shown in Table 5, even when the dataset size is reduced by 0.8 or even 0.6, the consistency method continues to provide a substantial performance

improvement, demonstrating that the proposed consistency approach remains effective even with limited data. This result underscores the robustness of our method. The findings suggest that our approach holds significant promise for scenarios where data availability is constrained, offering an effective solution to improve model generalization and performance even when working with smaller datasets. This insight could be particularly valuable in real-world applications where large-scale labeled data may not be easily accessible.

Table 5: Impact of dataset size of the CCL approach, where $K_r$ is the dataset reduction scaling factor.

| Method(Baseline) | OmniLabel | | | $D^3$ | | |
|---|---|---|---|---|---|---|
| | AP | AP-c | AP-d | FULL | PRES | ABS |
| GLIP-T | 19.3 | 23.6 | 16.4 | 19.1 | 18.3 | 21.5 |
| +ours($K_r$=0.6) | 27.2 | 30.4 | 24.7 | 26.2 | 25.2 | 29.3 |
| +ours($K_r$=0.8) | 30.1 | 34.1 | 27.4 | 28.9 | 28.1 | 31.3 |
| FIBER-B | 25.7 | 30.3 | 22.3 | 22.7 | 21.5 | 26.0 |
| +ours($K_r$=0.6) | 35.6 | 38.3 | 34.0 | 32.4 | 32.0 | 33.6 |
| +ours($K_r$=0.8) | 39.5 | 41.6 | 36.7 | 35.8 | 35.5 | 37.0 |

## B.3    CHOICE OF GENERATION SCHEME

**Additional Experiment on BBox Copy-Paste Method.**    To further explain our choice of generation scheme, we conduct an additional study comparing our pipeline to a standard copy-paste baseline. In this experiment, we use FIBER-B as the backbone and retain the same training schedule, and hyperparameters. The only difference lies in how foreground objects are extracted: instead of using SAM-generated masks, we directly use the bounding boxes from the training set and paste the entire box region onto new backgrounds. The results are shown in Table 6, in comparison with FIBER-B+ours from Table 1 of the main paper.

As seen in Table 6, the simple bbox copy-paste approach performs significantly worse than our full method. Although it does decouple foreground and background to some extent, the pasted bbox regions inevitably contain substantial non-foreground pixels, leading to impure representations. This degrades the model's ability to learn true background-invariant features—even with the added consistency loss—since the model tends to focus on the entire bbox region rather than the precise shape of the object.

**Discussion on Style-Transfer Methods.**    Regarding the alternative of style-transfer methods: these approaches typically require an additional style reference image and a dedicated style-transfer model (either CNN-based or diffusion-based) to generate stylized outputs, which introduces comparable or even greater complexity. More critically, most style-transfer methods alter not only the background but also distort the appearance of foreground objects, making them unsuitable for learning background-invariant representations. Some methods also require re-segmentation (e.g., using SAM) on the stylized images to recover object masks, which introduces noise and inconsistency—as discussed in Sect.B.6. This is especially unacceptable for object detection tasks, where precise annotations are essential.

Most importantly, typical style-transfer methods take both a content image and a style image as input, and may partially remove foreground objects during the transfer process. While some mask-aware methods attempt to mitigate this by treating the task as image inpainting, we have analyzed the theoretical limitations of applying inpainting-based techniques within our pipeline in the main paper, and provided supporting experimental results in Sect.B.6. These results show that inpainting methods perform poorly in our context.

## B.4    ANALYSIS ON CBDG SCENE DIVERSITY

While our current prompt set primarily focuses on Seasonal, Sky, and Natural Landscape categories, our goal is to begin with visually clean and semantically neutral backgrounds in order to isolate the effect of background variation on object-level consistency.

Table 6: Comparison Between Our Method and Simple BBox Copy-Paste Based on the FIBER Baseline

| Method | AP | Ap-c | Ap-d | FULL | PRES | ABS |
|---|---|---|---|---|---|---|
| FIBER-B+ours | 42.0 | 44.1 | 39.2 | 37.6 | 37.2 | 38.8 |
| FIBER-B + BBox copy-paste | 23.3 | 27.0 | 20.6 | 20.9 | 20.4 | 22.6 |

We conduct an experiment targeting urban, indoor, and architectural scenes. Specifically, we extract a subset of images from the $D^3$ test set that depict cityscapes and indoor urban environments. This subset includes the following categories: accidents, apples, auto salon girls, bakery, bananas, bedroom, boxing, cable car, cafe, chess, classroom, cooking, dining table, gymnastics, interview, kendo, library, living room, meeting, Olympic torch, origami, oven kitchen, pizza restaurant, refrigerator, restaurant, sandwich restaurant, snooker, street performers, street vendors food carts, study, supermarket cart, surgery hospital, toilet toothbrush, toy, traffic, and traffic accident, totaling 2,304 images.

We evaluate our model on this subset. As shown in Table 7, our method achieves results comparable to those on the full $D^3$ test set. This indicates that the learned representations generalize well even to previously less frequent scene types.

Table 7: Results on 2,304 $D^3$ Images from Cityscapes and Indoor Urban Environments

| Method | FULL | PRES | ABS |
|---|---|---|---|
| GLIP-T+ours | 29.7 | 28.9 | 32.1 |
| FIBER-B+ours | 37.5 | 37.1 | 38.6 |

## B.5 EXPERIMENTS ON PHRASE GROUNDING

To test whether our fine-tuning affects the baselines' object grounding capabilities, we revisit a key benchmark used in both the GLIP and FIBER papers: phrase grounding on the Flickr30K Entities dataset. This benchmark requires precise spatial localization of objects given natural language descriptions, and thus reflects the model's fine-grained visual grounding ability. We report Recall@K metrics for both GLIP-T and FIBER-B before and after applying our method:

Table 8: Recall@K metrics for both GLIP-T and FIBER-B before and after applying our method

| Model | Val | | | Test | | |
|---|---|---|---|---|---|---|
| | R@1 | R@5 | R@10 | R@1 | R@5 | R@10 |
| GLIP-T | 85.7 | 95.4 | 96.9 | 85.7 | 95.8 | 97.2 |
| GLIP-T+ours | 88.5 | 95.9 | 97.2 | 88.8 | 96.2 | 97.2 |
| FIBER-B | 87.1 | 96.1 | 97.4 | 87.4 | 96.4 | 97.6 |
| FIBER-B+ours | 89.9 | 96.6 | 97.7 | 90.0 | 96.8 | 97.7 |

The performance is overall better than the original baselines. Notably, GLIP-T + ours achieves consistent gains on R@1 for both validation and test sets, while FIBER-B + ours also improves across several metrics. These results confirm that our consistency-driven fine-tuning not only preserves but can even enhance the grounding capabilities of the baselines.

It is notable that this evaluation serves as a preliminary validation, since our model is currently fine-tuned on only a 250K subset of the training data, which is significantly smaller than the full datasets used to train GLIP and FIBER. We expect that with larger-scale training, our approach has strong potential to further surpass the baselines on phrase grounding benchmarks like Flickr30K Entities. In general, these additional experiments demonstrate that our method maintains or improves the original capabilities of the baselines while simultaneously enhancing robustness under domain shifts.

## B.6 Analysis on Inpainting Methods for Background Changing

In our background replacement approach, we initially adopt different methods. We explore models such as Glide Nichol et al. (2021) and Inpaint Anything Yu et al. (2023) to integrate the original image with newly generated backgrounds using image-to-image techniques. Our process involves providing three inputs to the image-to-image model: the original image, the bounding box of the foreground object as a prompt, and a simple textual description of the desired background. The model then selectively modifies the non-foreground regions of the image.

However, the results are unsatisfactory, presenting three major issues, as shown in Figure 7.a) Blurry Transition: the transition between the foreground and the generated background is often blurry, making the composite image appear unnatural. b) Extraneous Objects: the model tends to introduce extraneous objects into the background based on the foreground elements, which significantly disrupts the accuracy of subsequent object detection tasks. c) Logical Inconsistency: the generated images frequently contain logical inconsistencies that deviate from real-world plausibility, further diminishing their usability in our experiments.

We begin our investigation into synthetic data augmentation by utilizing the text-conditional inpainting capabilities of the GLIDE Nichol et al. (2021) model to modify image backgrounds according to textual prompts. However, initial experiments reveal that training with GLIDE-generated data leads to a notable drop in detection performance compared to the baseline, indicating limited utility of the inpainted samples. To improve data quality, we then experiment with the IAM Yu et al. (2023) model and introduce filtering strategies aimed at reducing noise and artifacts. It is worth noting that, unlike the experiments in the main paper, where we fine-tune with 0.25M images, here we used 0.09M images during this experiment to enable faster iteration at the early design stage. Consequently, the absolute performance numbers in Table 9 are lower than those in the main results, but the relative improvements between different generation methods remain valid. As presented in Table 9, even with filtering, the addition of IAM-generated data still results in a performance decline. Nevertheless, across all settings, our proposed CCLoss consistently demonstrates its effectiveness in enhancing model robustness.

Inpainting-based approaches such as GLIDE and IAM frequently produce visual artifacts or semantically inconsistent elements in the generated backgrounds. These imperfections compromise data quality and limit generalization during training. To overcome these issues, we transition to using the Stable Diffusion model Rombach et al. (2022) and adopt the full Context-aware Background Diversification and Generation (CBDG) pipeline detailed in our methodology. This revision yields a significant performance boost, affirming that high-quality synthetic data generated through our CBDG framework is critical for effective training and robust model behavior.

Table 9: Experiments on different generation method.

| Method(Baseline) | OmniLabel | | | $D^3$ | | |
|---|---|---|---|---|---|---|
| | AP | AP-c | AP-d | FULL | PRES | ABS |
| IAM(FIBER-B) | 20.5 | 24.6 | 18.0 | 18.4 | 17.8 | 20.3 |
| +ours(FIBER-B) | 26.1 | 31.2 | 22.2 | 23.5 | 22.8 | 26.1 |
| CBDG+ours(FIBER-B) | 32.4 | 34.6 | 29.4 | 30.3 | 30.0 | 31.4 |

## B.7 Progressive Design of the Intra-Modal Consistency Loss

The design of our intra-modal consistency loss is motivated by related work in self-supervised learning and domain adaptation, where preserving feature consistency under varying conditions has proven effective. In our framework, the development of the final loss function, CCLoss, follows a progressive and iterative design process. We begin by constructing a pairwise similarity matrix for the feature set $F = f_1, f_2, ..., f_n$ corresponds to the same object instance presented under different background conditions. In the initial formulation, we minimize the mean squared error (MSE) between the lower triangular part of this similarity matrix and an all-ones target matrix, encouraging uniform similarity among all feature representations. This variant is referred to as Matrix(FIBER-B) in Table 10. For each object in the same batch $n$, the loss is defined as follows:

$$\mathcal{L}_{\text{Matrix}} = \frac{2}{n(n+1)} \sum_{i=1}^{n} \sum_{j=1}^{i} \left( \mathbf{S}(f_i, f_j) - 1 \right)^2, \tag{8}$$

where $\mathbf{S}(f_i, f_j)$ represents the similarity between the feature vectors $f_i$ and $f_j$, which is computed using dot product.

To further enhance consistency, we incorporate the concept of variance reduction inspired by Variance Consistency Loss. Specifically, we minimize the variance among the features in $F$, promoting tighter feature clustering under intra-object background shifts. This second variant is denoted as Variance(FIBER-B) in Table 10. For each object in the same batch $n$, the loss is defined as follows:

$$\mathcal{L}_{\text{Variance}} = \frac{1}{n} \sum_{i=1}^{n} \| f_i - f_c \|^2, \tag{9}$$

where $f_c$ is the centroid of the image features for the c-th category object.

Building upon these two approaches, we finally adopt a contrastive learning framework to enforce discriminative yet consistent representation learning. This final version, termed CCLoss, outperforms the earlier variants and demonstrates superior robustness across all experimental benchmarks. It is worth noting that, unlike the experiments in the main paper, where we fine-tune with 0.25M images, here we used 0.09M images during this ablation to enable faster iteration at the early design stage. Consequently, the absolute performance numbers in Table 10 are lower than those in the main results, but the relative improvements between different loss functions remain valid. The progressive improvements reflected in Table 10 validate the effectiveness of each stage in the design of our loss function.

Table 10: Experiments on different consistency loss.

| Method(Baseline) | OmniLabel | | | $D^3$ | | |
|---|---|---|---|---|---|---|
| | AP | AP-c | AP-d | FULL | PRES | ABS |
| Matrix(FIBER-B) | 29.8 | 32.4 | 27.5 | 27.9 | 26.8 | 30.0 |
| Variance(FIBER-B) | 30.2 | 33.2 | 28.6 | 28.8 | 28.1 | 30.5 |
| CCLoss(FIBER-B) | 32.4 | 34.6 | 29.4 | 30.3 | 30.0 | 31.4 |

## C  RELATED WORK

### C.1  ROBUST OBJECT DETECTION

Robust object detection focuses on maintaining reliable performance under diverse challenging conditions, including occlusion, adverse weather, low-resolution inputs, domain shifts, and adversarial attacks. While traditional object detectors operate under idealized data assumptions, robust detection methods target the inherent variability of real-world environments. The field has seen significant progress through innovations in model architectures, training paradigms, and domain adaptation methodologies. Current research advances tackle robustness challenges through multiple directions: Adversarial training, where FROD Awais et al. (2023) improves detector robustness via modified backbones and lightweight components. Extreme condition adaptation, with UIA-YOLOv5 Ding et al. (2024) enhancing construction site detection through unified image adaptation. Noisy bounding box handling, as OA-MIL Liu et al. (2022) refines localization using classification guidance. Domain generalization, where Normalization Perturbation Fan et al. (2023) synthesizes feature styles for autonomous driving. Small object detection, with DenseNet-201 Akhtar et al. (2022) boosting YOLOv2 for traffic surveillance. Agricultural robustness, where smooth perturbations Mahmoud et al. (2024) improve YOLOv5 for root collar detection. These methods span adversarial resilience, environmental adaptability, and annotation noise tolerance.

## C.2 DATA AUGMENTATION FOR DETECTORS

Recent advancements in detector-specific data augmentation address limitations in classic methods like Mixup Zhang et al. (2017) and CutMix Yun et al. (2019), which primarily enhance diversity in fully supervised settings. Diffusion-based frameworks generate diverse contrail masks and scenes to improve detection robustness Lee & Yoo (2025), while attribution-driven methods leverage saliency maps to preserve critical features in low-level vision tasks Mi & Yang (2025). For few-shot object detection, MPAD Vu et al. (2025) integrates in-context object synthesis and hard sample generation via diffusion models. Nevertheless, persistent challenges—such as edge fidelity issues (e.g., blurred boundaries in diffusion-generated regions) and computational inefficiency—have led to renewed interest in hybrid augmentation strategies combining classical and modern techniques. Our proposed CBDG method uniquely integrates SAM (Segment Anything Model) and Stable Diffusion to generate semantically coherent, background-diverse images while minimizing label noise for consistency training. Unlike prior augmentation approaches, CBDG is designed to facilitate robust object-level feature learning across diverse environments in open-vocabulary scenarios, effectively mitigating common artifacts such as blurred boundaries.

## C.3 CROSS-MODAL OBJECT DETECTION MODELS

Cross-modal object detection develops rapidly with advances in vision–language pretraining and grounding. Early frameworks such as CLIP Radford et al. (2021) and ALIGN Jia et al. (2021) establish the foundation by learning scalable image–text representations, which inspire a series of detection-oriented extensions. MDETR Kamath et al. (2021) aligns objects and textual queries through a transformer-based design, while OWL-ViT Minderer et al. (2022) enables open-vocabulary detection with vision transformers. Subsequent models, including G-DINO Liu et al. (2024) and OFA-DOD Xie et al. (2023), further advance grounding accuracy and generalization across diverse tasks. Among these, GLIP Li et al. (2022b) and FIBER Dou et al. (2022) emerge as representative and widely adopted frameworks. GLIP introduces a scalable region-level vision–language pretraining formulation, whereas FIBER adopts a two-stage coarse-to-fine strategy that improves alignment at both image and region levels. Both models provide strong baselines for language-supervised object detection, support text-based grounding tasks, and are publicly available, which makes them ideal foundations for evaluating and extending consistency-based improvements. Their popularity and architectural compatibility also highlight the model-agnostic nature of our proposed approach.

## C.4 SURVEY OF EXISTING BACKGROUND DATASETS

To assess the suitability of existing datasets for our method, we conduct a detailed survey of publicly available background datasets. We find that nearly all of them contain varying degrees of foreground objects, which fundamentally conflicts with our training objective: pasting finely segmented foregrounds (via SAM) onto clean backgrounds to enforce background-invariance.

**Foreground contamination.** Many background datasets include objects that should not appear in background-only scenes. For instance, when sampling 200 random images from BG-20k Li et al. (2022a)—often cited as a background-only dataset—we found that 36% contained recognizable foregrounds such as roses, dolphins, butterflies, and other clearly foreground elements. Other datasets (e.g., SUN09 Xiao et al. (2010), Stanford Background Gould et al. (2009), Cityscapes Cordts et al. (2016)) often include pedestrians, vehicles, or other labeled objects, introducing semantic confusion. This undermines the consistency objective by entangling background pixels with foreground semantics.

**Domain-specificity.** Some datasets (e.g., NH-HAZE Ancuti et al. (2020), RESIDE Li et al. (2018)) are restricted to specific conditions such as hazy weather, making them unsuitable for general-purpose training.

**Scale limitations.** Many datasets are too small to support large-scale pretraining or fine-tuning. For example, the Stanford Background Dataset contains only 715 images, Cityscapes provides 2,975 training images within a single urban domain, and NWPU VHR-10 has merely 150 background images.

For methods that explicitly separate foreground and background, dataset purity is crucial. Existing background datasets either suffer from foreground contamination, domain bias, or insufficient scale, and therefore cannot fully meet the requirements of our approach.

# D  CBDG

## D.1  PROMPT TO LLM FOR BACKGROUND GENERATION

Listing 1: Example prompts for generating natural landscape background images with seasonal variations, time of day, and weather conditions.

```
Generate a list of diverse prompts for background image generation,
    ↪ focusing on natural landscapes (sky and atmosphere) with seasonal
    ↪ variations. Provide varied descriptions covering different times of
    ↪  day, weather conditions, and visual elements. The prompts should
    ↪ be clear, creative, and specific for use in diffusion models.

Generated examples:
1. A vibrant spring meadow with tall grass and colorful wildflowers,
    ↪ where bees and butterflies flit from bloom to bloom under the warm
    ↪ sun.
2. A frozen river cutting through a snowy valley, with ice floes floating
    ↪  on the surface and snow-capped trees lining the banks.
3. A tropical waterfall plunging down into a crystal-clear pool,
    ↪ surrounded by dense jungle and mist rising from the water below.
4. A golden autumn forest at sunset, with leaves in shades of red and
    ↪ orange covering the ground and a crisp chill in the air.
5. A stormy summer beach scene with dark clouds overhead, waves crashing
    ↪ against the shore, and seagulls fighting the strong winds.
6. A misty morning in a bamboo forest, with soft sunlight filtering
    ↪ through the tall green stalks and dew on the leaves.
7. A panoramic view of rolling hills under a twilight sky, with the first
    ↪  stars appearing and a crescent moon rising.
8. A desert oasis at high noon, with palm trees surrounding a small pool
    ↪ of water and heat waves distorting the distant horizon.
9. A cherry blossom grove in full bloom, with pink petals floating on a
    ↪ gentle breeze and covering a quiet pathway.
10. A northern lights display over a frozen lake, with vibrant green and
    ↪ purple streaks reflected in the icy surface.
```

We employ three distinct prompts to generate background descriptions for three different categories using ChatGPT. While the templates for these prompts are identical, they vary slightly in terms of the generated content, as illustrated in Listing 1. The primary difference lies in the type of content generated, tailored to each category.

## D.2  EFFECT OF IoU THRESHOLDING ON TRAINING QUALITY

To address the issue of incomplete or inaccurate foreground object segmentation by SAM, which is unacceptable for downstream object detection tasks, we implement a post-processing step to filter out problematic segmentation results. Specifically, we calculate the bounding box (BBox) of each segmented mask and compare it with the corresponding ground truth BBox by computing the Intersection over Union (IoU). Only segmentation results with an IoU greater than a predefined threshold $T_{IoU}$ are retained; otherwise, they are discarded. The threshold $T_{IoU}$ we take is 0.75.

This post-processing step mitigates the impact of noisy BBox labels caused by potentially inaccurate SAM-generated masks, thereby improving the quality of training data and enhancing model performance.

## D.3  ANALYSIS ON THE PROCESSING PROCEDURES OF DIFFERENT DATASETS

In our CBDG process, we apply two different augmentation strategies to our combined dataset of Flickr30k and Objects365. For images with more than one class in them, we leverage the prior

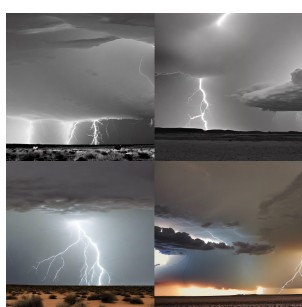 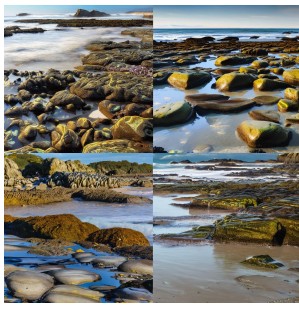 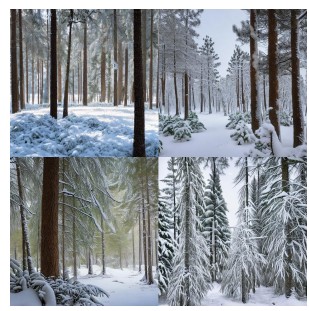

| (a) Type: Sky | (b) Type: Landscape | (c) Type: Season |
|---|---|---|
| Prompt: A thunderstorm sky with dark, rolling clouds and the occasional flash of lightning, set against a vast, empty desert. | Prompt: A rocky coastline at low tide, with tide pools exposed and seaweed-covered rocks, as the waves slowly return. | Prompt: A serene forest of snow-covered pine trees, with a light dusting of snow falling from the branches and the ground blanketed in white. |

Figure 6: Three types of background images we generate. Given a prompt from LLM, we use stable diffusion to generate random background images.

knowledge of category labels and bounding box information to directly extract the foreground objects using the SAM model. Once the foreground is segmented, we proceed to replace the background, ensuring that the core object remains intact while the surrounding context is altered. This method allows us to generate diverse augmented images while maintaining object consistency.

For images with only one class in them, we begin by augmenting the number of object categories within the images, expanding the variety and complexity of the scenes. After this, we apply the same background replacement technique as used for the Flickr30k dataset, using the extracted foreground objects and swapping the background accordingly. This dual augmentation approach helps us enrich the dataset, enhancing its diversity and challenging the model to adapt to a wider range of visual contexts. By utilizing both the category label and bbox priors in conjunction with the SAM model, we can effectively augment the datasets in a way that introduces meaningful variation while preserving object relevance, which is crucial for improving model robustness and generalization across different scenarios.

In the process of category augmentation, to minimize the impact of dataset size, the augmented objects are exclusively selected from 0.22M images from the Objects365 dataset, excluding objects from the images we did not select within the original Objects365 dataset.

### D.4 ANALYSIS ON BACKGROUND GENERATION

As shown in Figure 6, the background images we generate are both rich and diverse, with significant variation in elements such as scene types, lighting conditions, weather changes and texture details. These images are created by providing prompts derived from various language descriptions generated by a large language model (LLM) to the Stable Diffusion model. The prompts are designed to capture different scenes, ensuring that the generated backgrounds exhibit a high degree of diversity.

There are three main categories of background images: seasonal, sky, and natural landscape. In total, we have 13,185 unique descriptions, resulting in 144,654 generated images. The breakdown of categories and the corresponding number of images is as follows: seasonal (3387 descriptions, 48,156 images), sky (3399 descriptions, 48,210 images), and natural landscape (3399 descriptions, 48,288 images). As shown in Figure 6, even for the same language description, the generated images demonstrate a strong diversity in terms of their visual appearance, making each one unique in its own right.

This diversity is key to enhancing the quality and variability of the generated datasets, allowing us to better capture the complexity of real-world scenes and improve the robustness of subsequent tasks, such as OVOD and REC. Moreover, the diversity in the backgrounds plays a crucial role in supporting the consistency constraints we impose on the images. Unlike traditional data augmentation methods, which may lead to homogenized backgrounds, our approach ensures that the model is exposed to a

wide range of backgrounds across different contexts. This enhances the model's robustness, making it more adaptable and effective in diverse scenarios, and ultimately improving its performance on tasks that require consistency across varying environments.

### D.5 ANALYSIS ON IMAGE SELECTION AND CATEGORICAL AUGMENTATION

For the Objects365 dataset, we initially select 0.22M images, ensuring that all categories are covered to support more comprehensive model training and evaluation. During the data selection process, we track the categories and the corresponding number of objects (bounding boxes) in each image. To ensure category balance, we first analyze the distribution of categories across the dataset and set a minimum coverage requirement, ensuring that each category is represented by at least one image.

Subsequently, we introduce a priority strategy based on category diversity and object count to sort and select the images. The core idea behind this priority strategy is to prioritize images that contain a larger number of categories while favoring those with a higher number of objects. This approach allows for more efficient category coverage within a limited dataset and ensures that the dataset has an adequate number of objects to support robust object detection model training.

For each image, we define its priority score as:

$$S(I) = N_{\text{bbox}}(I) + N_{\text{cat}}(I) \tag{10}$$

where $S(I)$ represents the priority score of image $I$, $N_{\text{bbox}}(I)$ denotes the number of objects (bounding boxes) in image $I$, $N_{\text{cat}}(I)$ refers to the number of categories in image $I$.

We sort the images by priority score and gradually select them. While ensuring category coverage, we further optimize the data selection based on object count and category diversity, such that the final subset contains both rich category information and reasonable object density. This strategy significantly enhances the representativeness of the dataset, providing more challenging and generalizable training data for subsequent object detection tasks.

### D.6 PSEUDO CODE OF CBDG

The overall process of contextual bootstrapping data generation is summarized in the Algorithm 1.

---

**Algorithm 1** Contextual Bootstrapping Data Generation

---

**Require:**
 1: $I$: Original image with objects and bounding boxes.
 2: $D_{fg}$: External dataset for object augmentation.
 3: $N$: Number of potential placement positions.
 4: $N_R$: Maximum resizing attempts.
 5: $\alpha$: Resizing factor.
 6: Themes: Background themes (*Seasonal, Sky, Natural Landscape*).
**Ensure:**
 7: $I^*$: Augmented image with diverse objects and backgrounds.
 8: Extract object using SAM $\mathcal{S}$: foreground objects $\leftarrow \mathcal{S}(I)$.
 9: **for** each object in $I$ **do**
10:    Randomly select an object $o$ from $D_{fg}$.
11:    Attempt placement at $P$ positions:
12:        $(x_{o_k}, y_{o_k}) \in P \backslash (x_o, y_o), o_k \in O_{i \notin C}$.
13:    category augmentation: $I' \leftarrow I$
14:    **if** no valid placement found **then**
15:        Resize $o$: $o \leftarrow o \times 1/\alpha$.
16:        Repeat placement attempts up to $N_R$ times.
17:        **if** still no valid placement **then**
18:            Skip and select another image.
19:        **end if**
20:    **end if**
21: **end for**
22: Generate background prompts using LLM $\mathcal{G}$:
23:    $t' = \mathcal{G}(t)$.
24: Generate background images using Stable Diffusion $\mathcal{D}$:
25:    $b = \mathcal{D}(t')$.
26: Replace background in $I'$:
27:    Extract foreground and compose new image: $I^* = \mathcal{S}(I'_{box}) \oplus b, b \in D_{bg}$.
28:
29: **return** $I^*$.

---

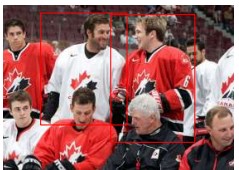 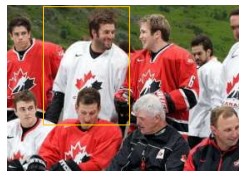 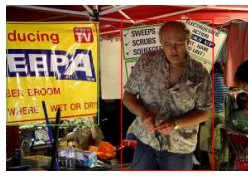 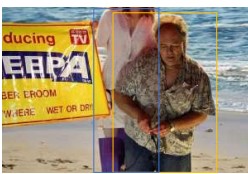

Caption: Two members of a Canadian hockey team are talking to each other while the rest of the team sit or stand around them.

Problem:
Blurry Transition: The hair is blurry.

Caption: A man in a button down short-sleeved shirt testing a product being advertised by the yellow and white signs.

Problem:
a) Blurry Transition: The hair and the shirt is blurry.
b) Extraneous Objects: The woman beside the man does not exist in the original image.

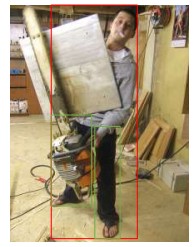 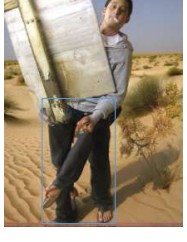 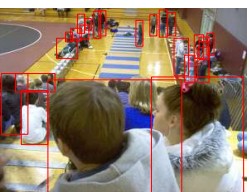 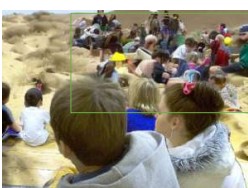

Caption: A young man holds up a piece of wood and a chainsaw as presses a cigarette between his lips.

Problem:
Logical Inconsistency: The person has three lips.

Caption: People are sitting in bleachers watching some activity below them.

Problem:
Extraneous Objects: The inpainting model generates many people.

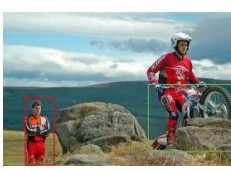 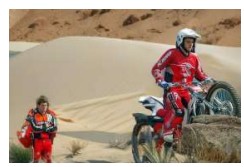 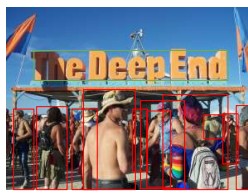 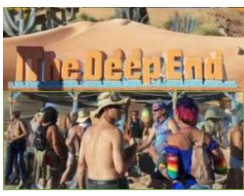

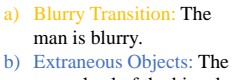

Caption: A man is looking on as another man attempts to climb a small boulder with his dirt bike.

Problem:
a) Blurry Transition: The man is blurry.
b) Extraneous Objects: The rear wheel of the bicycle.

Caption: A group of people stand in front of an establishment called The Deep End.

Problem:
Extraneous Objects: The inpainting model generates many people.

Figure 7: Failure cases of background replacement using inpainting method Yu et al. (2023). Each case consists of paired images: the left image shows the original input, while the right image demonstrates the replaced background with corresponding artifacts. Cases are presented in a two-per-row layout for comparison.