# OpenReview forum: "Consistency Beyond Contrast: Enhancing Object Detection Robustness through Scene-Augmented Feature Alignment"
_ICLR.cc/2026/Conference — ICLR 2026 Conference Withdrawn Submission_

### Official Review · Reviewer_BB3x · 2025-10-30

**Soundness:** 2
**Presentation:** 3
**Contribution:** 3
**Rating:** 4
**Confidence:** 4

**Summary:**

This paper introduces Contextual Consistency Learning (CCL), a method to improve robustness in open-vocabulary object detection by enforcing consistency of object features across different backgrounds. It combines Contextual Bootstrapped Data Generation (CBDG), which creates synthetic training pairs with the same object in varied scenes using SAM and Stable Diffusion, and a Contextual Consistency Loss (CCLoss) to align feature representations. The approach shows performance gains on OmniLabel and D3 benchmarks, and shows  resilience to background shifts.

**Strengths:**

S1. The work identifies an important overlooked issue: current open-vocabulary detectors can overfit to training backgrounds, causing inconsistent detections when context changes. This focus on enforcing intra-modal (vision-only) consistency is novel in a field that has mostly prioritized cross-modal (vision-language) alignment. The motivation is well-argued – improving robustness by making object features invariant to environmental changes addresses a gap not sufficiently explored in prior OVOD works.

S2. The proposed CCL framework is design is model-agnostic, with two complementary parts. CBDG is a reasonable data augmentation pipeline and leverages existing models - SAM to accurately isolate foreground objects and Stable Diffusion to generate new background scenes. The makes method simple yet flexible as: it can be integrated into any open-vocabulary detector’s training regime without architectural changes.

S3. The empirical improvements are noticeable. On two open-vocabulary detection benchmarks, OmniLabel and D3, the CCL-enhanced models dramatically outperform their non-CCL counterparts, though robustness limited to background.

S4. CCL being simple and efficient, adding no inference-time cost or latency overhead. This is a practical strength – the model’s deployment speed and complexity are unchanged, which is important for real-world adoption.

**Weaknesses:**

W1. Limited Robustness Evaluation - The robustness analysis, while focused on backgrounds, is not as comprehensive as it could be. The paper considers only one type of distribution shift – changes in background scenery. Yet robustness in vision can be challenged by many other factors: lighting conditions, weather or camera effects, partial occlusion of objects, viewpoint changes, adversarial noise, etc. One of the study Chhipa et al. (2024) [1] benchmarked open-vocabulary detectors on a range of perturbations (the COCO-C corruption suite and COCO-O occlusion/occclusion-like scenarios) and found that even state-of-the-art models suffer significant performance drops under various image degradations. By focusing narrowly on background variation, the submission stops short of demonstrating that CCL improves general robustness. It will be helpful to discuss and possibly evaluated suggested challenges in robustness. It remains unclear whether the gains would carry over to other shift types.

[1] Chhipa, P. C., De, K., Chippa, M. S., Saini, R., & Liwicki, M. (2024, September). Open-Vocabulary Object Detectors: Robustness Challenges Under Distribution Shifts. In European Conference on Computer Vision (pp. 62-79).

W2. The success of CBDG hinges on the quality of two processes: foreground segmentation and background image generation. Any errors in these could propagate into training data and potentially confuse the model. However, the paper does not report any quantitative metrics or analysis of these aspects. This would be another essential to strengthen the paper.

W3. The submission does not compare against or cite some recent works that also tackle robustness in detection, which could situate its contributions better. For example, ADD (Attribution-Driven Data Augmentation) Mi et al. (2025) [2] proposes a saliency-based augmentation strategy to boost model performance by focusing on important image regions and mitigating the influence of irrelevant background pixels. While ADD was developed in the context of image super-resolution but the core idea – using Calibrated Attribution Maps to guide augmentation so that critical content is preserved – is conceptually related to what CCL is trying to achieve.

[2] Mi, Z. Y., & Yang, Y. B. (2025). ADD: Attribution-Driven Data Augmentation Framework for Boosting Image Super-Resolution. In Proceedings of the Computer Vision and Pattern Recognition Conference (pp. 23101-23110).

 W4. Equations (5–7) define the loss, there is no clear justification or analysis of why category centroids and cosine similarity across backgrounds ensure semantic consistency.

W5. The paper mentions “diverse backgrounds” from three prompt groups (Seasonal, Sky, Natural Landscape) but provides no diversity statistics or qualitative failure rate of the generative pipeline.

**Questions:**

Weakness section covers the questions.

---

### Official Review · Reviewer_r9gp · 2025-10-31

**Soundness:** 3
**Presentation:** 3
**Contribution:** 3
**Rating:** 4
**Confidence:** 3

**Summary:**

This paper presents Contextual Consistency Learning (CCL), a framework that aims to improve robustness in open-vocabulary detection and grounding tasks by enforcing context-invariant feature learning. The approach combines two components: (1) Contextual Bootstrapped Data Generation (CBDG), which uses SAM to segment objects and composites them onto diffusion-generated backgrounds guided by LLM prompts; and (2) a Contextual Consistency Loss (CCLoss), which aligns regional features across varied backgrounds to encourage intra-modal invariance. CCL integrates seamlessly into existing detectors like GLIP-T and FIBER-B, requiring only one epoch of fine-tuning on a joint dataset (~0.25M real images plus 144k generated scenes). The method reports strong gains on OmniLabel and D3 benchmarks (FIBER: +16.3 AP and +14.9 AP; GLIP: +12.9 AP and +10.9 AP) and introduces a new robustness protocol, D3BC, that tests performance under synthetic background replacement. While results convincingly show improved contextual robustness, the heavy reliance on synthetic augmentations and missing hyperparameter details limit the clarity and generalizability of the conclusions.

**Strengths:**

The new D3BC benchmark is well-aligned with the paper’s goal, systematically testing contextual robustness through independent background replacements. The dataset must be public.

Ablations effectively separate the roles of data augmentation (CBDG) and the loss (CCLoss), showing complementary contributions.

The one-epoch fine-tuning setup demonstrates efficiency compared to large-scale retraining typical in this field.

strong empirical results. The large gains on both OmniLabel and D3 benchmarks show consistent improvements in robustness and generalization.

**Weaknesses:**

Some of the important hyperparameters are missing from the main text. Without them, reproducibility and sensitivity analysis remain incomplete.

**Questions:**

Please follow the strengths and weaknesses.

---

### Official Review · Reviewer_s2qi · 2025-10-31

**Soundness:** 3
**Presentation:** 3
**Contribution:** 2
**Rating:** 4
**Confidence:** 4

**Summary:**

This paper shows that ope-vocabulary object detection (OVD) models are not robust against background changes. It proposes a framework (CCL) to improve the robustness of OVD models. CCL includes two main components: (1) data augmentation component (CBDG) and (2) contextual consistency objective (CCLoss). CBDG uses SAM to identify objects within images and uses stable diffusion to construct images by augmenting foreground objects (using SAM) and diversifying backgrounds using background description (generated by ChatGPT) according three categories: seasonal, sky, and natural landscapes.   CCLoss optimizes a image-based and language-based contrastive objective among images with same foregrounds but different backgrounds to enforce learning robust representation invariance to the backgrounds.

CCL is implemented on two popular OVD models (i.e. GLIP and FIBER). Particularly, it shows  that OVD models cannot detect same objects when appeared in different context (i.e. background). Then it shows that CCL can acheive higher performance on both evaluation datasets (i.e. D^3 and OmniLabel) and it outperforms OVD baselines on background augmented images. The ablation study shows that both components (i.e. CBDG and CCLoss) are effective to improve robustness and performance of OVD models on D^3 and OmniLabel benchmarks.

**Strengths:**

1- Paper studies an important limitation in current OVD models. Specifically, as these models are intended to be adopted in different scenarios, it’s important to adopt robustness towards different context when employed in real-world scenarios
2- Experimental results show that the proposed framework can significantly improve the baselines (i.e., before finetuning) on popular benchmarks such as OmniLabel.
3- Ablation study shows the effectiveness of each component. i.e., additional augmented data through CBDG can improve the performance, but the best results are achieved when fine-tuned with contrastive loss.

**Weaknesses:**

1- **Lack of discussion on DA/DG methods, specifically ones that incorporate OVD methods.** The paper does not discuss domain adaptation or domain generalization methods, especially those that incorporate open-vocabulary detection (OVD). The focus is on improving generalization to unseen environment, but a comparison or discussion with OVD-related DG approaches would help clarify contribution and effectiveness of the paper

2- the background diversity is similar/identical through CBDG method, which does not effectively show the robustness to **unseen background/environment** as claimed in the paper (e.g. L093). Some additional analysis could help: e.g.,  1) split the background prompts into training and test across all categories, and/or (2) keep one category as a held-out set. i.e. train on two (e.g. Seasonal and sky) evaluate on landscape. How does the model's robustness compare to baselines and data only version? (i.e. CCL without CLLoss)


3- While D3 and Omnilabel are not part of the training data (i.e. some background variations) paper doesn’t provide any quantifiable evaluation on how model is performing on real-world unseen background, despite the claim that baselines struggle under varying real-world conditions (e.g. L078). Could the background diversity in evaluation datasets and training datasets be further clarified? Or alternatively, it might be helpful t evaluate CCL on existing DG benchmarks such as BDD100k and real-to-artistic benchmarks (e.g. CLIPART, WATERCOLOR, CLIPART). However, one caveat with the latter is that foreground objects are also impacted by domain shift, which is somewhat a more realistic setting than just changing the background, like images shown in Fig.4 don’t seem realistic (e.g. baby in a suitcase on water), but maybe out of the context of this study.

4- **Some additional ablations are needed and/or are helpful**
(1) Effectiveness of the language vs. image-based components. This is especially important since the language branch is not applicable to GLIP-like architectures, as mentioned in the paper.
(2) Effect of batch size or global centroid quality in consistency computation. The quality of centroid features (e.g., in Eq.6 and 7) depends on the batch size. So such an ablation is needed to better understand the impact of batch size and the quality of centroid. One could test the method with different batch size and/or use EMA average throughout training

5- While CBDG also inserts objects (to single object images) and diversifies backgrounds, the contrastive training is conducted on same foreground but different backgrounds. Hence, it’s not clear why adding more foreground objects is needed? And how it impact the generalization?

6- L448-451 claims  "…..models incorporating our CCL approach demonstrate significantly improved robustness with much smaller performance drops." However, comparing results in Table 1 and Table 2, shows that FIBER-B baselines performance drops less than FIBER-B + CCL. For instance on FULL the performance drops  by 0.11% (22.7 -> 20.1) and 0.12% (37.6 -> 33.1) on FIBER-B baseline and FIBER-B + CCL. Clarifying this could be helpful to better understand the effectiveness and interpretation.

**Questions:**

1- Clarify the difference between prompt tokens t_c and t_ck for the same category c. Are the prompt different and how would that impact Eq.7?



2- [Minor/suggestion] Add baseline + data (i.e. without CCLoss) in the tables as it’s a more fair baselines. I think its minor because results is already provided in Table 3 which shows effectiveness of both modules.


3- [Minor] It seems like consistency loss is over objects-level features. However, prior works shows that both image-level and object-level consistency can further improve the performance. How is the results differ? [1]

4 - In table 5 in supp, K_r is used to ablate over data sizes. How’s this ablation performed? fewer augmented images are generated? smaller amount of samples from original data is chosen?, or after generation of synthetic data a subset (e.g., 0.6) of data is selected?  Also, if a subset of the data after generation is selected, the selected subset may still contain all original images (i.e. just less augmentations) Would this be fair?

5- I couldn’t find the results for GLIDE?
6- Also, How to ensure non-overlapping foreground augmentation? i.e. if multiple objects are added does the method ensure the added objects don’t overlap?

---

### Note · Authors · 2025-11-12

I have read and agree with the venue's withdrawal policy on behalf of myself and my co-authors.